

# Effects of reflective warning markers on wildlife

Yi-Hao Fang[1,2,3,4,5,*], Ying Gao[1,2,3,4,5,*], Yin Yang[1,2,3,4,6], Kun Tan[1,2,3,4], Yan-Peng Li[1,2,3,4], Guo-Peng Ren[1,2,3,4], Zhi-Pang Huang[1,2,3,4], Liang-Wei Cui[2,3,4,5] and Wen Xiao[1,2,3,4]

[1] Institute of Eastern-Himalaya Biodiversity Research, Dali University, Dali, Yunnan, China
[2] Collaborative Innovation Center for Biodiversity and Conservation in the Three Parallel Rivers Region of China, Dali University, Dali, Yunnan, China
[3] The Provincial Innovation Team of Biodiversity Conservation and Utility of the Three Parallel Rivers Region from Dali University, Dali University, Dali, Yunnan, China
[4] The Key Laboratory of Yunnan Education Department on Er'hai Catchment Conservation and Sustainable Development, Dali University, Dali, Yunnan, China
[5] Faculty of Biodiversity Conservation and Utilization, Southwest Forestry University, Kunming, Yunnan, China
[6] School of Archaeology & Anthropology, Australian National University, Canberra, Australia
[*] These authors contributed equally to this work.

## ABSTRACT

Light pollution has become one of the top issues in environmental pollution, especially concerning how secondary light pollution, such as from traffic reflective materials, influences animal distribution and behavior. In this study, 15 camera traps were set up at sites with or without reflective warning markers (RWM) in coniferous forests on Cangshan Mountain located in Dali Prefecture, China. The results showed that the number of independent photographs and species at sites without RWMs were significantly higher than those at sites with RWMs. Significant differences were found between daytime and nighttime composition of bird species and non-flying mammals between two sites. This study found that RWMs had negative effects on wildlife, with the avoidance response of birds to RWMs being more obvious than that of animals at daytime. It is recommended that the use of reflective materials be carefully considered, especially in protected areas.

## INTRODUCTION

Artificial structures are having a growing influence on biodiversity (*Pechmann et al., 1991*; *Soga et al., 2014*). The development of cities and roads is considered as one of the most important factors influencing the survival of wildlife, often with complex effects (*Sauvajot et al., 1998*; *Seto, Güneralp & Hutyra, 2012*; *Bennie et al., 2015*) such as altering wildlife habitats, increasing habitat fragmentation, directly damaging wildlife, and indirectly affecting wildlife by changing ambient sounds, light and odors (*McLellan & Shackleton, 1988*; *Trombulak & Frissell, 2000*; *Forman & Deblinger, 2000*; *Coffin, 2007*; *Bolger et al., 2008*; *Summers, Cunnington & Fahrig, 2011*; *Arevalo & Newhard, 2011*). Light pollution

Corresponding authors
Liang-Wei Cui,
cuilw@eastern-himalaya.cn
Wen Xiao,
xiaow@eastern-himalaya.cn

from road lighting systems is considered as one of the most serious environmental interferences (*Hölker et al., 2010*), with 35% to 50% of light pollution being generated from the traffic lighting systems. This has become an increasingly important environmental problem (*Stoilova & Stoilov, 1998*; *Lyytimäki, 2013*). Reflective warning markers (RWMs) have been widely used on highways, smaller roads, in urban areas (*Wolshon, Degeyter & Swargam, 2002*), and other areas that are being developed due to expanding urbanization and road construction. In addition, various types of RWMs are frequently used in agriculture, horticulture (*Layne, Jiang & Rushing, 2002*), architectural engineering (*Li et al., 2008*), and even in protected areas (*Reinius & Fredman, 2007*; *Wu, Zhang & Zou, 2007*).

How RWMs influences wildlife has been rarely considered. The materials used in RWMs are not self-luminous and directs people with conspicuous colors in the daytime and by reflection at night. Conspicuous colors, such as red, are commonly considered to be aposematic, or natural warning signals, and are used by insects and birds to signal their undesirability to potential predators (*Stevens, 2007*; *Myczko et al., 2015*; *Iniesta, Ratton & Guerra, 2017*). Although the intensity of reflected light at nighttime is low intensity but noticeable, it is still regarded as a special source of light pollution (defined as secondary light pollution). Similar reflector materials are sometimes used to alter the behavior off animals and direct their movement away from roads to reduce wildlife-vehicle collisions (*Anke, Peter & Torsten, 2018*). However, there is still a large gap in the research of physiological and ecological effects of night illumination on animals (*Gaston & Bennie, 2014*).

A goal of this study was to understand how RWMs influence wildlife. By comparing animal species with independent photographs shot by the cameras grouped at stations with and without RWMs, this study assessed how nighttime reflection and a daytime warning color affected wild animal species. The results will provide guidance on the future use of RWMs.

## MATERIALS & METHODS

### Study area and camera trapping

Cangshan Mountain, located in Dali Prefecture of Yunnan Province, China (Fig. 1A), is in the Cangshan Mountain and Erhai Lake Nature Reserve. Previous surveys have shown Cangshan Mountain is rich in native flora and fauna (*Shen, 1998*; *Zhao et al., 2017*). The study site is located between Mocan and Heilong streams ($25°36'-25°40'$N, $100°06'-100°11'$E, altitude range 2,550–2,800 m, Fig. 1A). Vegetation at the site has a ground cover of 40%–80% and is dominated by 15–20 m tall Armand pine (*Pinus armandii*). Human activities are strictly controlled in this nature reserve and almost no artificial light enters the study area.

To avoid impacts caused by environmental heterogeneity, five sub-regions were selected on a single slope of a stream divide. From April 9, 2017 to July 23, 2017, 30 camera traps were set up in study stations. Within each sub-region, three stations included RWMs (RWM group), and another three stations were without RWMs (control group) (Fig. 1A). The distances between the nearest two camera traps were 100–200 m. The RWMs were

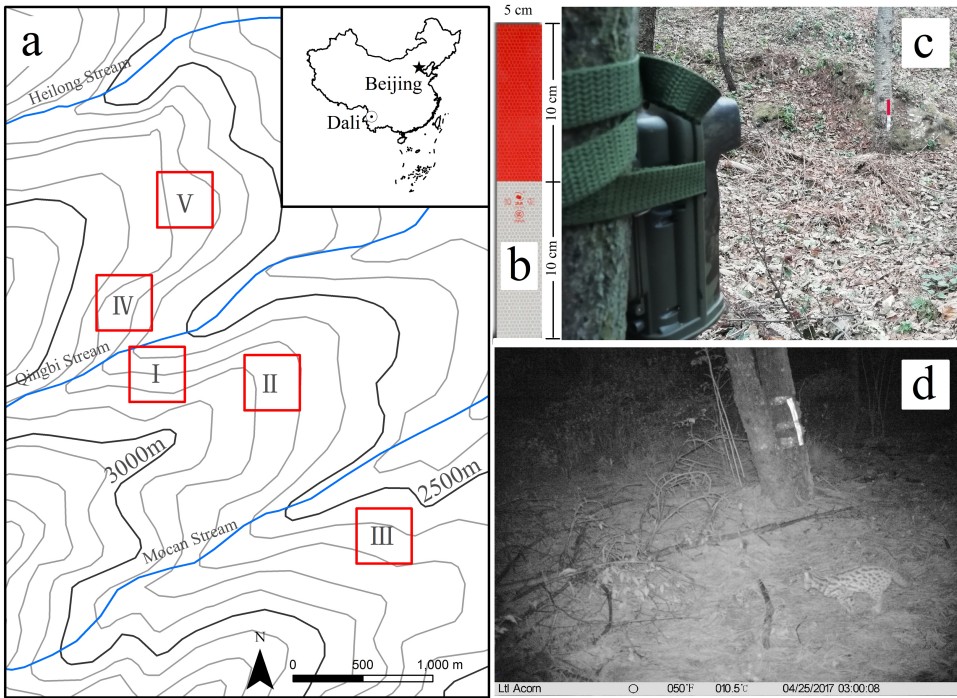

**Figure 1** **Study area and the camera trap with a reflective warning marker.** (A) The location and stretch map of the study site. (B) Reflective warning marker (RWM). (C) A camera trap and the RWM installation, (D) A leopard cat captured moving in front of the RWM by a camera trap.

installed 4 m in front of the camera traps and 0.3 m aboveground (Fig. 1C). The size of the RWMs were 20 cm × 5 cm and were designed with 10 cm × 5 cm red colored strips and 10 cm × 5 cm white colored strips. Both strips reflect light but are not self-luminous (Fig. 1B).

Ltl Acorn® 6310MC cameras were used and set at medium sensitivity for motion, with two photos and a 10 s video captured in rapid succession upon each trigger. Camera installation and basic photographing methods followed specifications proposed by *Xiao et al. (2014)*. All cameras made by Zhuhai Ltl Acorn Electronics Co., LTD in Zhuhai City of Guangdong province, China.

## Data processing and statistical analysis

When the camera traps were collected, all memory cards and data were processed using software CTIMCS1.1 (a software for manually sorting camera trap photographs that was developed independently by the Institute of Eastern-Himalaya Biodiversity Research, Dali university, http://www.eastern-himalaya.com.cn/contents/3/990.html). Animals on the photos were identified to the species level with the time and date being noted. The identification of mammal species was accomplished using *Smith et al. (2010)* and bird species were identified using *MacKinnon et al. (2000)*. Nocturnal mammals such as rodents and bats were excluded from the analysis because they could not be accurately identified on the photographs captured at night. *O'Brien, Kinnaird & Wibisono (2003)* used several
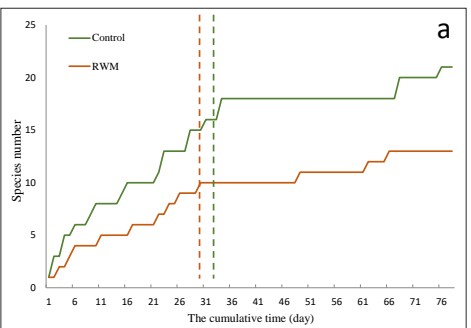
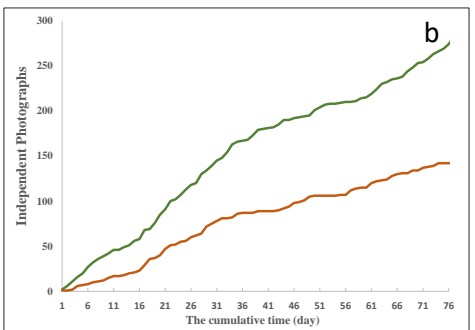

**Figure 2  Cumulative curves of species number and independent photographs.** (A) The cumulative curves of species number in RWM group (brown line) and control group (green line); the dotted line show when 75% of species were recorded the corresponding group. (B) The cumulative curves of independent photographs in RWM group (brown line) and control group (green line).

independent photographs to estimate species abundance and activities. We used the same method to uniquely define each photograph in a series consecutive photos taken of the same or different species, or of the same individuals taken during 30 min. Nighttime was defined as lasting from 19:30 to 6:30, and daytime lasted from 6:30 to 19:30.

To assess if RWMs significantly changed the diversity and abundance of species, we used a general linear mixed model, with the number of species and independent photographs captured separately on each camera as the dependent variables (*Bolker et al., 2009*). We considered the sub-regions as random-effects, and the presence or absence of RWM as the fix-effects. We conducted a covariance analysis to test if the slope of the independent acumination line between the two groups was significantly different.

To estimate the species composition between the groups, we used the Jaccard index (*Jaccard, 2010*), which is a community similarity index calculated as $q = c/(a + b - c)$, where $q$ is the Jaccard index, $a$ and $b$ are the number of species in each group, and $c$ is the number of common species in both groups. Indices between: $0 \leqq q \leqq 0.25$ were considered extremely dissimilar, $0.25 < q \leqq 0.50$ were moderately dissimilar, $0.50 < q \leqq 0.75$ were moderately similar, and $0.75 < q \leqq 1$ were extremely similar. All analyses were completed with Software R (*R Core Team, 2018*).

# RESULTS

In this study, the 30 cameras effectively worked for 76 days and recorded 25 wild animal species belonging to 14 families, 11 of which were mammal species and 14 were bird species (Appendix S1). More than 75% of the species were recorded in the first 30 days in the RWM group and in the first 31 days in the control group, indicating that the sampling effort was adequate (Fig. 2A).

## Number of species and independent photos

The camera traps captured 13 (RWM group) and 21 (control group) species. The linear mixed model analysis showed that the cameras with RWMs recorded significantly higher

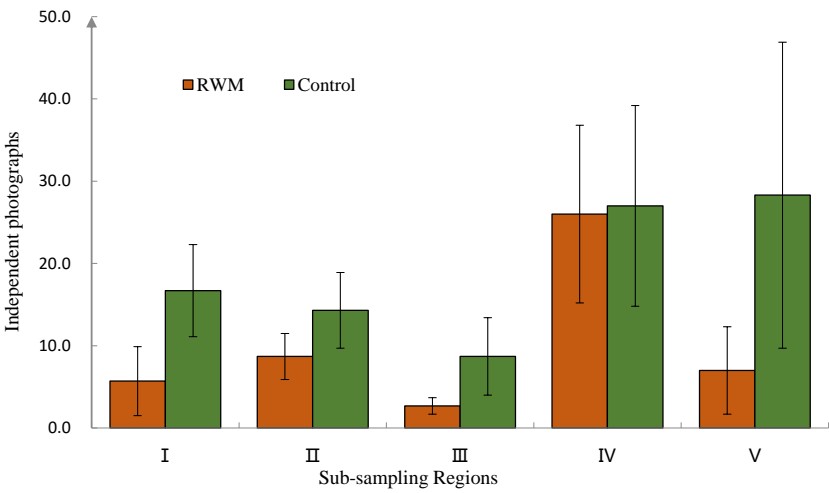

**Figure 3 Independent photographs of sub-regions.** Independent photographs number of RWM group (brown, mean ± SD) and control group (green) in the five sub-regions (I, II, III, IV, V).

species number than those without ($2.60 \pm 1.99$ vs. $4.93 \pm 2.99$, $t = -2.70$, $p = 0.012$, Appendix S2).

Cameras traps with RWMs captured 146 independent photographs, whereas the control group had 284 photographs (Fig. 2B). Although not significantly different, cameras with RWMs captured fewer independent photographs than did cameras without RWMs ($10.00 \pm 13.52$ vs. $20.13 \pm 21.30$, $t = 1.66$, $p = 0.130$, Appendix S2). Stations with RWMs had a lower number of independent photos than did the control group in all five sub-sampling regions, indicating these regions had a lower animal density (Fig. 3). The slopes of the number of independent photographs by dates for the two groups were significantly different ($df = 152$, $t = 20.60$, $p < 0.001$).

## Composition of community species

Among the 25 species of animals captured by camera traps, seven of the mammalian species and one avian species was recorded in both treatments (Fig. 4). The Jaccard index of species for both treatments was 32%, which was classified as moderately dissimilar. However, the bird species composition was extremely dissimilar between two groups (Jaccard = 7.6%), and the mammal species composition was moderately similar (Jaccard = 58.3%).

More species and independent photographs were captured in the daytime than in the nighttime for both groups (Fig. 5). Eleven bird species and four mammal species were only captured in the daytime, and one bird species and five mammal species were only captured at night. Two mammal species, the long-tailed goral (*Naemorhedus caudatus*) and leopard cats (*Prionailurus bengalensis*), were captured during daytime and nighttime in both groups (Fig. 6).

Among the species captured in the daytime, long-tailed thrushes (*Zoothera dixoni*) and moustached laughingthrushes (*Garrulx cineraceus*) only appeared in areas with RWMs. Ten other bird species only appeared in the areas without RWMs. One bird species, the Lady
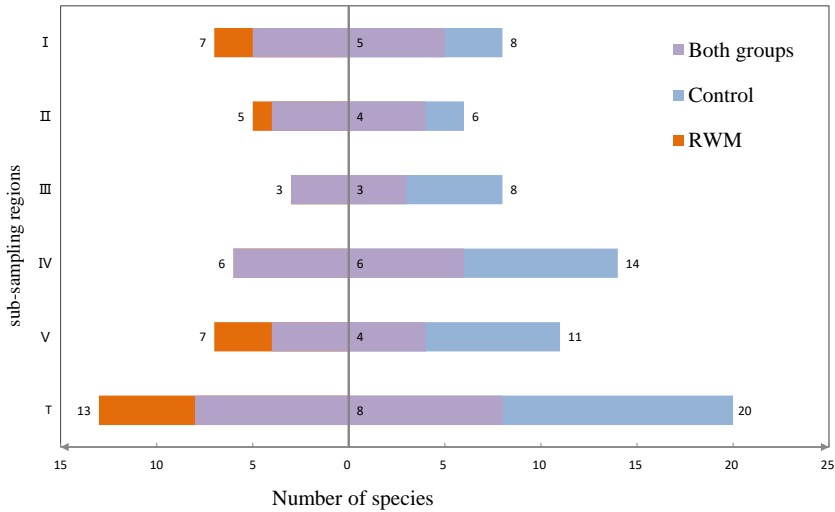

**Figure 4 Species number captured in the sub-regions.** The species numbers captured in both group (purple), only in the RWM group (orange) and control group (blue) in the five regions (I, II, III, IV, V) and all regions (T).

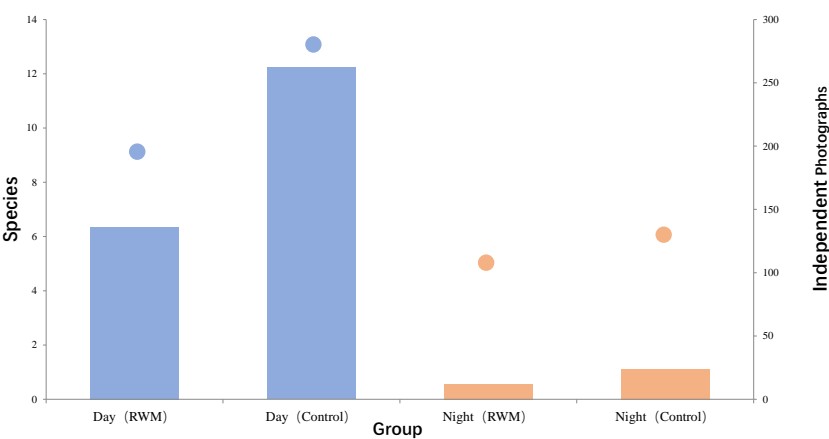

**Figure 5 Species number and Independent photographs number captured in the day and the night.** Species number (column) and Independent photographs number (dot) captured in RWM (blue) and Control (brown) groups during the day and the night.

Amherst's pheasant (*Chrvsolophus amherstiae*), and four mammal species were captured in both groups (Fig. 6).

Tawny owls (*Strix aluco*), Yunnan hares (*Lepus comus*), and Malayan porcupines (*Hystrix brachyura*) were only captured at night in the stations with RWMs, whereas masked palm civets (*Paguma larvata*) and red muntjacs (*Muntiacus muntjak*) only appeared in areas without RWMs.
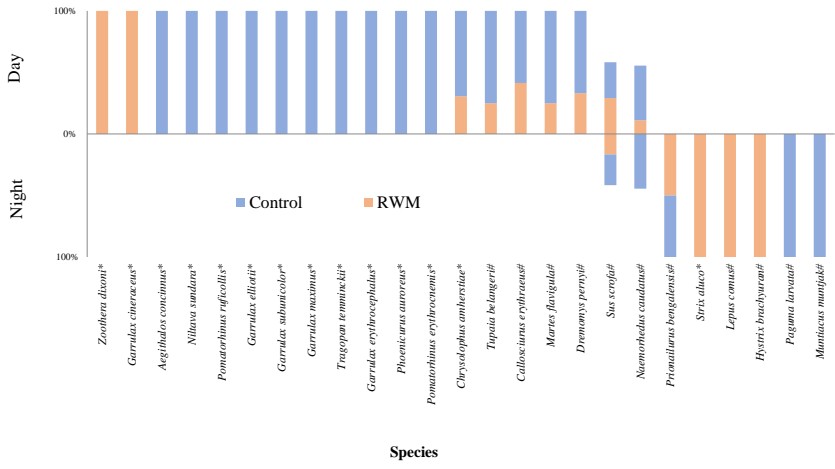

**Figure 6** **The percentage of independent photographs of species in the day and night.** The percentage of independent photographs of species recorded in RWM (brown) and Control (blue) groups in the day-time (up) and nigttime (down)* bird # mammal.

## DISCUSSION

The effects of light pollution on wildlife have been widely documented (*Cabrera-Cruz, Smolinsky & Buler, 2018*; *McLaren et al., 2018*). *Jones & Francis (2003)* dramatically reduced the mortality of migratory birds by decreasing the intensity and power of light beams. *Adrien et al. (2018)* claimed that artificial lights, primarily from highways and associated traffic, were continuously increasing in protected areas and biodiversity hotspots around the world. In our study, artificial lights were rare and nearly all reflected light originated from the moon and sun. Our study areas could be different from other areas that experience strong artificial light. However, the present study strongly suggests that light pollution during daytime and RWMs secondary light pollution during nighttime may influence wildlife distribution.

### Effects of RWMs on wildlife during nighttime

Although the number of species during nighttime differed slightly between the camera groups with and groups without RWMs, the number of independent photographs in the stations with RWMs was nearly 50% lower than the control group. The differences might be related to the different density or activities pattern of the species recorded in both groups. Of the eight species captured at night, tawny owls, Yunnan hares, and Malayan porcupines were only found in the areas with RWMs. Tawny owls are a nocturnal predatory species that preys mainly on small rodents. The short-eared owl (*Asio flammeus*) is another owl that has been found to have better predation efficiency at night. Consequently its prey, deer mice (*Peromyscus maniculatus*), have decreased their foraging behavior and other activities to avoid detection in highlighted environments (*Navara & Nelson, 2007*). Therefore, the materials might be attractive to the owls because they might use light reflected by RWMs during hunting.

The three species that only appeared in the areas without RWMs were primarily prey species in an ecosystem. Their predation avoidance abilities may keep them from appearing in places with RWMs that may assist predators. Leopard cats and wild boars appeared in both groups, perhaps because they are highly adaptable and live around human residential areas. The leopard cats hunt near villages and wild boars search for food in farmlands (*Rajaratnam et al., 2007*; *Schley & Roper, 2003*).

### Effects of RWMs on wildlife in the daytime

During the daytime, there were considerable differences in the number of species and independent photographs captured between the two groups. The stations with RWMs had nearly 30% fewer species and 50% fewer independent photographs. The daytime differences in species composition were probably caused by the different community composition of birds. Seven babblers captured, of which six were in the control group and only one was found the in RWM group. White-bellied pheasants were the only species that showed no avoidance response in the RWM groups during daytime. Red colored RWMs can increase visual stimulation in a manner similar to the alert color of insects (*Stevens, 2007*; *Iniesta, Ratton & Guerra, 2017*). A bird-repelling device has invented and functions effectively by using a red material that reflects light in an intermittent pattern (*Doty III, Turkewitz & Byers, 2003*).

## CONCLUSIONS

Results of this study suggest that RWMs, as crucial road components, can significantly affect the distribution of wildlife. Thus, it is necessary to consider the environmental effects of RWMs in assessing road construction. At present, some countries and regions have introduced standards and policies on night lighting and regulations to control light pollution (*Cha et al., 2014*; *Raap, Pinxten & Eens, 2015*; *Ryu & Lee, 2015*; *Lyytimäki, 2013*); Presently there are no policies on the use of the RWMs.

We propose that specifications be added to regulations on the use of reflective warning signs for ecological important sites, such as protected areas and national parks. The use of reflective warning signs should be carefully considered, including the plastic and glass materials that have similar reflection and warning effects. The use of RWMs should be minimized in important biological corridors that cross highways to reduce traffic interference with wildlife migration. Concurrently, installing RWMs in traffic corridors may prevent wild animals from moving onto highways and consequently reduce the rate of traffic accidents involving animals (*Reed, Woodard & Pojar, 1975*). The avoidance response of wildlife to RWMs can also be used for repelling bird in orchards, farmlands, and feed mills. Color signals can alert birds to danger and lower the frequency of accidents with non-reflective or non-glare clear glass (*LeMessurier, 1986*; *Doty III, Turkewitz & Byers, 2003*; *Klem, 2009*). Wind turbine blades and power-supply cables can be made with red and white warning colors that can protect birds from flying into fan blades (*Wei et al., 2011*). In sum, wildlife responses to RWMs can be used for improving wildlife management; however, more studies are needed to better understand how RWMs influence wildlife.

## ACKNOWLEDGEMENTS

Thanks Mr. Zhu Xiao-Ming for his help in writing the article.

### Funding

This study was supported by the National Natural Science Foundation of China (NO. 31560599, NO. 31560118, NO. 31860164 and NO. 31860168), the Program for Backup Talents of Young Academic and Technical Leaders in Yunnan Province (2015HB047), the Yunnan provincial program of 10,000 intellects - leading intellects for industrial technologies (Cui Liang-Wei, 2018), and the China Green Foundation. The funders had no role in study design, data collection and analysis, decision to publish, or preparation of the manuscript.

### Grant Disclosures

The following grant information was disclosed by the authors:
National Natural Science Foundation of China: 31560599, 31560118, 31860164, 31860168.
Program for Backup Talents of Young Academic and Technical Leaders in Yunnan Province: 2015HB047.
Yunnan provincial program of 10,000 intellects - leading intellects for industrial technologies.
China Green Foundation.

### Competing Interests

The authors declare there are no competing interests.

### Author Contributions

- Yi-Hao Fang conceived and designed the experiments, performed the experiments, analyzed the data, contributed reagents/materials/analysis tools, prepared figures and/or tables, authored or reviewed drafts of the paper, approved the final draft.
- Ying Gao analyzed the data, prepared figures and/or tables, authored or reviewed drafts of the paper.
- Yin Yang authored or reviewed drafts of the paper, approved the final draft.
- Kun Tan analyzed the data, prepared figures and/or tables, authored or reviewed drafts of the paper, approved the final draft.
- Yan-Peng Li contributed reagents/materials/analysis tools, authored or reviewed drafts of the paper.
- Guo-Peng Ren analyzed the data, authored or reviewed drafts of the paper.
- Zhi-Pang Huang conceived and designed the experiments, contributed reagents/materials/analysis tools, authored or reviewed drafts of the paper.
- Liang-Wei Cui conceived and designed the experiments, authored or reviewed drafts of the paper, approved the final draft.

- Wen Xiao conceived and designed the experiments, performed the experiments, analyzed the data, contributed reagents/materials/analysis tools, authored or reviewed drafts of the paper, approved the final draft.

## Data Availability

The raw measurements are available in the Supplemental Files.

## Supplemental Information

Supplemental information for this article can be found online at http://dx.doi.org/10.7717/peerj.7614#supplemental-information.

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
