# Peer review of "Effects of reflective warning markers on wildlife"

_PeerJ, doi:10.7717/peerj.7614_

## Round 0.1 · original submission · Major Revisions

Please follow the reviewer comments and revise your manuscript accordingly or address the comments in your rebuttal. When you have made all the corrections to your revised manuscript, I strongly suggest that you have the manuscript edited by a native English speaker. I will not accept a revised manuscript that is not clear and grammatically correct.

Reviewer 1 ·

Basic reporting

Very poor writing which hinders the comprehension of the manuscript. Inconsistent terminology and unclear/confusing relationship between the objectives, metrics described in the Methods and their usage in the Results (including Figures).

Experimental design

The study seems well design but important details about the characteristics of the sampling area are missing; which ultimately can hinder the comparison between the sites with and without RWM.

Validity of the findings

Discussion of the results in a fairly light way: 1) mainly based on speculations (with limited support from the existing literature), 2) without any discussion about the levels of reflected light in the study area, and 3) no mention to possible study limitations/ confounding factors, which may have impacted on study results and/or compromise the endorse of such strong practice recommendations by the authors.

Additional comments

GENERAL COMMENTS
This paper addresses an interesting topic: the effect that traffic warning markers may have on wildlife. The study was conducted in a mountain area in China, with cameratraps being placed in 5 sub-sampling areas to assess the relative abundance, species richness and diversity of birds and non-flying mammals in the presence (treatment) or not (control) of a reflective warning marker. There is extensive literature about effectiveness of wildlife warning reflectors (exclusively developed for altering wildlife behavior nearby roads and reducing wildlife–vehicle collisions); however, studies about the effect that regular traffic reflectors have on animals seem to be uncommon.

However, the manuscript will need major revisions in order to achieve the quality and scientific level expected for a publication. In my opinion, the most important shortfalls that need to be addressed are:
• Poor writing which makes the comprehension of the ms difficult. A revision by a native English speaker is strongly recommended.
• Problems with ms structure which hampers its clarity and impact (i.e. help guiding future research and/or the implementation of mitigation measures). E.g. formal “Materials & methods” and “Conclusion / recommendation” sections are missing.
• Inconsistent terminology and poor interconnection between objectives/question of the study, the metrics described in the Methods, and the results presented in the different Results sub-sections. For example, methods describe two relative-abundance metrics (UCR and CR) which usage is unclear in the results (including Figures 3, 4 and 6).
• Lack of detail in the description of methods and results (e.g. poor description of study area; captions of figures very incomplete/unclear).
• Discussion of the results in a fairly light way, without any mention to 1) the levels of reflected light in the study area, or 2) possible study limitations/ confounding factors; which may have a impact on study findings and/or compromise the endorsement of such strong practice recommendations.

For more details, please check the specific comments given below.

SPECIFIC COMMENTS

Line 1 – This is a very broad title that might mislead the readers. It would be important to make it more informative (e.g. “Effects of reflective warning markers on wildlife: a case study from Cangshan
Mountain, China”).

Line 23 – besides diversity, the study also assessed species richness and relative-abundance, correct?.

Line 23 and 28 - Replace “animals” by “mammals” or, to be more precise, by “non-flying mammals”, since bats were not included in the surveys.

Line 32 - Is should be numbered (1. Introduction).

Line 48 – “It is expected to further study on whether…”- difficult to understand (please revise grammar). Is this the first study about the effect of traffic warning markers or not?

Line 40 – delete the blank space between the parenthesis and the final period. Revise all ms accordingly.
Line 61-63 – This is somehow confusing as the reader does not know what are “independent photographs”. Revise the sentence to clarify the reader about which parameters were assessed (relative-abundance, species richness and diversity?). It is also important to keep the terminology consistent across the different sections of the ms.

Line 65 – It would be helpful to create a formal section dedicated to the Methods (as recommended by PeerJ standards and manuscript template), which included a sub-section with the description of the study area (2.1).

Line 66-69 & Fig. 1a – A more informative map (also with scale and north) would be helpful. Perhaps, keep the same broad map (which shows the location of Dali region) but then add a second map, showing the Cangshan mountain, the Mocan and Heilong streams, and the location of 5 sub-areas (where the camera were placed). See chkeck the Figure 1 from O’Brien et al. 2003.

Line 70-73 – Provide more details about the characteristics of the study area, namely in terms of 1) shrub cover, as it can affect the amount of reflected light that reaches the animals; and 2) existence of roads (or other sources of artificial light) nearby. Since the RWM are not self-luminous, it would be important to understand what are the “expected” levels of light reflected and what is its primary source (only moon light and/or artificial lights).

Line 73 – were placed?

Line 74-80 – Information about cameratraps and their settings a part of “Data collection” and, therefore, should be moved to the next section. Alternatively, you may rename this section, for example, "2.1 Study area and camera trapping" and the next one "2.2 Data processing and statistical analysis".

Line 83 – Please add a reference for the software M-photov1.0.

Line 84-85 - delete the text box (with a letter "d") on top of the main text.

Line 85-86 – Replace “rodent rat” by “rodents”. Besides rodents, which nocturnal animals were also excluded?

Line 87 - Replace “beasts” by “species”.

Line 90-93 – These sentences are a transcription from O'Brien et al. (2003). Revise the text in order to use it in own words or, at least, make it clear that this is the criteria defined/used by O'Brien et al. (2003).

Line 93-95 – To which “two relative-abundance indices” are you referring? UCR and CR? If that is the case, move the sentence in between (“Night-time was defined ... setting time point the sun”), in order to make it clear. Please also note that, as far as I can understand, UCR and CR are not the same two relative-abundance indices described by O'Brien et al. (2003). The way the text is currently written may be misleading for the readers.

Line 96-102 – the meaning of each formula would be easier to understand if the definition of the parameters was done immediately after each formula. E.g.
Unit capture rate (UCR) and capture rate (CR) were calculated by the following formulas:
(1)UCR=n/t×100%
where n was the overall number of photos captured by a single camera, and t was the effective (shooting)working days that were obtained by subtracting the days of camera maintenance from the total days during the study period.
(2)CR=N/T×100%
where N was the total number of photos by all working cameras, and T was the total effective working days.

Line 101- Replace “shooting” by “ working” in order to keep the terminology consistent.

Line 104-105 – It would be important to clarify that the “two communities” you are referring to, are the ones present in RWM and control sites (to avoid confusion with the bird and mammal communities).

Line 112 – and the left?

Line 119 / - Include a reference to support the claim “The number of independent photographs is a reliable indicator of animal distribution density.”. Please also make sure that terminology is used consistently (e.g. “animal distribution density” Versus “relative abundance index”).

Line 119-122, Figure 3 and 4 – Its not clear what are the metrics being used in each situation (e.g. 10 ± 3.4 vs. 20.1 ± 5.3). When are UCR and CR used?

Line 101- Replace “RWM treat” by “RWM treatment”.

Line 125 – in different places?

Line 126 - The results of the Jaccard index would be the same if they were calculated separately for birds and mammals?

Line 131-132 - make it clear that you are referring to the difference between day and night times (otherwise it makes no sense to repeat here the same results of section 3.1).

Line 133-135 – hard to understand. Please revise the sentence grammar.

Line 140 – “leopard cats and wild roars” – correct to “boars”. Latin names of both species are also missing.

Line 150-153 – It is not completely clear why do you think that animals’ abundance was nearly 50% less in RWM sites (compared to controls). The “different features of RMW between the day and the night” are clear, but, during night-time, do you believe that the amount of light reflected is enough to explain such strong avoidance effect? This seems, to some extent, implausible taken in consideration that even the effectiveness of wildlife warning reflectors (which objective is exclusively to deter wildlife from approaching roads) is not completely clear (e.g. Benten et al., 2018). Thus, it would be important to discuss the study results bearing in mind the known (or “expected”) levels of reflected light that reaches the animals in this particular study site (given the presence roads, shrubs, etc.), how it may differ (or not) from other study areas, and the results of studies where the effectiveness of wildlife warning reflectors was tested.
Benten, A., Annighöfer, P., Vor, T., 2018. Wildlife Warning Reflectors’ Potential to Mitigate Wildlife-Vehicle Collisions—A Review on the Evaluation Methods. Front. Ecol. Evol. 6, 37. https://doi.org/10.3389/fevo.2018.00037

Line 154-165 –The authors speculate (based on a study on owls and their better predation efficiency under moonlight) that reflective materials may help nocturnal hunters. However, it seems implausible that the amount of light reflected by RWM is enough to create such opportunistic behaviour in owls, hares and porcupines (and the opposite effect on preys). Once again, it would be important to discuss the results considering the known (or “expected”) levels of light reflected in this particular study site.

Line 175-192 – The impact of the study results could be improved through the conversion of the current 2nd-level section (4.3 Suggestion on RWM usage) into a new 1st-level section (e.g. 5. Conclusion and recommendations).

Line 190-191 – The extrapolation of the recommendation for the wind energy and power lines context seems too much. I advise you to delete this sentence or to revise properly the extensive literature (on mitigation measures of bird and bat collision with wind turbine and overhead wire) before making any recommendation on this matter.

Figures - Revise the captions of all figures to make them accurate and more informative. All metrics and abbreviations need to be clearly defined and consistent with the terminology used across the ms.
Figure 3 –“Effective independent photos number” - which metric is this? “Number of independent photos/ p working day”?
Figure 4 – “Effective Independent photo number” – do you mean ”Accumulated number of independent photos”?
Figure 5 – replace the label of X-axis by “Number of species”
Figure 6 – replace the label of Y-axis by “Number of species”.
Figure 7- hard to understand. It would be helpful to included the name of the species directly in the x-axis (instead of using numbers) and to identify which are birds and which are mammals (to help readers not familiar with China's native fauna).

Appendix 1 - Correct the spelling of “protection level”. Please clarify to which reference you are referring to and the meaning of level ”II”. Also include in the caption of the figure the meaning of CITES (including the meaning of “II” and “III”).

Reviewer 2 ·

Basic reporting

English needs substantial correction


The work by Kamiel Spoelstra and colleagues is very relevant to this study and should be included (intro/discussion) https://nioo.knaw.nl/en/light-nature

Experimental design

I think there are 5 treatment sites and 5 control sites (paired)? Please make explicit

Maybe I missed it, but was there a light source during the night that was applied to the reflective strips? Please clarify, especially if there was none, because along roads car’s headlights would periodically illuminate the reflective treatment.

Validity of the findings

x

·

Basic reporting

The English language should be revised and improved throughout the article to ensure that an international audience can clearly understand the text. This might be done by a native English. I give some examples:

-These phrases should be rewritten to make them more understandable: L26-28, L35, L48, L51, L56-58, L84-86, L107, L116-118, L124-125, L131, L133-136, L144-148, L154-156, L176

-There are also orthographic errors in the use of punctuation marks: L41, L43, L63, L132, L159, L163, L180.

Experimental design

This paper presents data on the effects of artificial and reflective warning-signals on biodiversity, a kind of light pollution with relevance to biological conservation. Such works, linking human activities with their potential impacts on biodiversity, are needed to understand the responses of wildlife to pollution and also to define proper conservation strategies. The paper suits well to the aims and scope of PeerJ.

Field methods and design are appropriate according to the study objectives.

There are some inconsistences when using the initials of reflective warning markers (RWM). Throughout the article the authors use “RMW” many times in place of “RWM”. See for example L22-27.

According to the design of the work it would be clearer to define “control” and “experimental” (or “RWM”) sites or cameras.

Probably the observed effect of RWM is related to the smell of these markers. Could authors reject this hypothesis?

Validity of the findings

Some analyses need more information or could be improved.
For example according to the study design, it would be more appropriate to use t-test for dependent samples (pairs of control and experimental cameras) better than t-test for independent samples (L107-108).
L108. More information is needed about “proportional test” or alternatively give a reference.
L1016-118. It seems authors have performed a species richness analysis but no information is given about this kind of analysis. Did authors use specific software (for example “EstimateS”)? How did authors estimate the total richness of species)? Did authors calculate the percentage of species observed (an indicative of the sampling effort)?
L121. You might use a factorial ANOVA to test the effects of RWM and regions in the number of photos.
L121-122. You might use an analysis of covariance or a Homogeneity-of-slopes model to test for differences in the rate of accumulation of photos.
L131-132. You might use a factorial ANOVA to test the effects of RWM and period of the day in the number of species.

Additional comments

Some parts of the article could be improved to make it more clear and attractive. Please find my comments below.

L23. I guess you mean “birds and mammals” and not “birds and animals”.
L26. “Jaccard” and not “Jaccad”

L51. I guess authors are referring to “natural warning colors”. Please, include “natural”
L53. What do you mean by “destructive colors”?
L53-54. I guess authors are referring to aposematic signals.
L56-60. These sentences are not clear. Are you referring to artificial or natural warning colors?

L71-75. Then authors placed 3 control-cameras and 3 RWM-cameras in each of the 5 areas (sub-regions). Please clarify.
L91. Please explain what you mean by “consecutive”.
L102. Please clarify what you mean by “camera maintenance”.

L113. Please explain “second-class protected wildlife” or alternatively give a reference.
L1016-118. It seems authors have performed a species richness analysis but no information is given about this kind of analysis. Did authors use specific software (for example “EstimateS”)? How did authors estimate the total richness of species)? Did authors calculate the percentage of species observed (an indicative of the sampling effort)?
L121. You might use a factorial ANOVA to test the effects of RWM and regions in the number of photos.
L121-122. You might use an analysis of covariance or a Homogeneity-of-slopes model to test for differences in the rate of accumulation of photos.
L131-132. You might use a factorial ANOVA to test the effects of RWM and period of day in the number of species.
L140. “Wild roars”?
L151-153. I do not understand this sentence. Please clarify.
L154. I guess you mean “species” and not “animals”. Please clarify.
L159. Why “Reflective materials may help nocturnal hunters like owls at night”?
L201-203. Please include the name of the journal.
L256-260. Raap et al. should be placed after Pechmann et al.

Figure 7. Please, separate “Birds” and “Mammals” in the text.
Table 1. This table needs more information. For example columns “protection leve” and “CTIES” need to be defined.

---

## Round 0.2 · Minor Revisions

There are still some concerns with the grammar. Your manuscript is novel, and we would like to see it published. However, the readability and clarity need improvement. Please consider these additional reviewer comments in revising your manuscript.

·

Basic reporting

1. Language should be corrected by a native Englishman. I am not a native speaker but some sentences have grammatical errors and look strange, even for me. Corrections made by authors alone are not enough.

2. Figures should be corrected. Night and day parts of graphs should be marked somewhere. Please, remove any raw numbers (e.g. means with sd). This can be read from axes. Keeping them is very misleading.

3. Raw data are hard to understand. It is a large Excel file with many sheets. Actually, I could not find any column with plot ID in order to repeat GLMM. There is no description of the variables. Species names are not in a tidy format. Scientific (Latin) names should be separated from Chinese. Moreover, it would be more useful to see English names instead of Chinese ones. Remove any graphs. Explain every column. Sheet names should be clear.


Minor comments
Please, change „Cameras traps” to „Camera traps” throughout the manuscript (e.g. lines 132, 13;5).

Line 99 „We identify…” should be „We identified…”

Lines 132-134. A vague sentence. „Significantly” what? Less or more? Please, check the order of the figures in the brackets.

Line 136: Please insert hypo after 284.

Lines 146- 147: Please change into „…while the species compositions were extremely dissimilar (Jaccard =7.6%) or moderately dissimilar (Jaccard =58.3%) between the two groups in birds and mammals, respectively.”

Line 148: Change „Independent” into „independent”

Lines 186-187: Change the entire sentence into „Therefore, the materials might be attractive to the owls because they might use light reflected by RWMs during hunting.”

Experimental design

The methodology seems to be fairly good. I liked using mixed models. However, for count data more natural choice is the use of generalized linear mixed models with Poisson error and log-link function instead of the linear mixed model. Please, check excellent guidance in papers published in PeerJ:
- Harrison XA. 2014. Using observation-level random effects to model overdispersion in count data in ecology and evolution. PeerJ 2:e616 https://doi.org/10.7717/peerj.616
- Harrison XA, Donaldson L, Correa-Cano ME, Evans J, Fisher DN, Goodwin CED, Robinson BS, Hodgson DJ, Inger R. 2018. A brief introduction to mixed effects modeling and multi-model inference in ecology. PeerJ 6:e4794 https://doi.org/10.7717/peerj.4794
Moreover, there is no need to use ANOVA. You can use a generalized linear mixed model again. Just introduce interaction term between time and treatment to the model.

Validity of the findings

Findings are novel and can be a substantial contribution to ecology, namely the effects of light pollution on animal communities. Better statistical tests could strengthen the impact of the manuscript as it was mentioned in point 2 above.

Additional comments

The manuscript was substantially improved as compared to the earlier version. The strong side of this manuscript is a novelty. However, it still requires major revision before it can be accepted in the international journal.

---

## Round 0.3 · accepted · Accept

Thank you for your efforts to improve the readability of your manuscript.